# A Review of Conventional and Novel Treatments for Osteoporotic Hip Replacements

**DOI:** 10.3390/bioengineering10020161

**Published:** 2023-01-25

**Authors:** Fahad Alabdah, Adel Alshammari, Araida Hidalgo-Bastida, Glen Cooper

**Affiliations:** 1Engineering College, University of Hail, Hail 55476, Saudi Arabia; 2School of Engineering, University of Manchester, Oxford Road, Manchester M13 9PL, UK; 3Department of Life Sciences, Faculty of Science & Engineering, Manchester Metropolitan University, Manchester M15 6BH, UK

**Keywords:** osteoporosis, hydrogels, total hip replacement, tissue scaffolds

## Abstract

Introduction: Osteoporosis is a skeletal disease that severely affects the mechanical properties of bone. It increases the porosity of cancellous bone and reduces the resistance to fractures. It has been reported in 2009 that there are approximately 500 million osteoporotic patients worldwide. Patients who suffer fractures due to fragility cost the National Healthcare Systems in the United Kingdom £4.4 billion in 2018, in Europe €56 billion in 2019, and in the United States $57 billion in 2018. Thus, osteoporosis is problematic for both patients and healthcare systems. Aim: This review is conducted for the purpose of presenting and discussing all articles introducing or investigating treatment solutions for osteoporotic patients undergoing total hip replacement. Methods: Searches were implemented using three databases, namely Scopus, PubMed, and Web of Science to extract all relevant articles. Predetermined eligibility criteria were used to exclude articles out of the scope of the study. Results: 29 articles out of 183 articles were included in this review. These articles were organised into three sections: (i) biomechanical properties and structure of osteoporotic bones, (ii) hip implant optimisations, and (iii) drug, cells, and bio-activators delivery through hydrogels. Discussion: The findings of this review suggest that diagnostic tools and measurements are crucial for understanding the characteristics of osteoporosis in general and for setting patient-specific treatment plans. It was also found that attempts to overcome complications associated with osteoporosis included design optimisation of the hip implant; however, only short-term success was reported, while the long-term stability of implants was compromised by the progressive nature of osteoporosis. Finally, it was also found that targeting implantation sites with cells, drugs, and growth factors has been outworked using hydrogels, where promising results have been reported regarding enhanced osteointegration and inhibited bacterial and osteoclastic activities. Conclusions: These results may encourage investigations that explore the effects of these impregnated hydrogels on osteoporotic bones beyond metallic scaffolds and implants.

## 1. Introduction

Osteoporosis is a skeletal disease that severely affects the mechanical properties of bone. It increases the porosity of cancellous bone and reduces the resistance to fractures [1]. It has been reported that there are approximately 500 million osteoporotic patients worldwide [2]. Patients who suffer fractures due to fragility cost healthcare systems around £4.4 billion in the United Kingdom [3], €56 billion in Europe [4], and $57 billion in the United States [5], annually. Thus, osteoporosis is problematic for both patients and healthcare systems.

The presence of osteoporosis often compromises the success of using orthopaedic devices due to the influence of reduced bone quality on stability, and secondary fractures [6,7]. The lack of mechanical stability results in aseptic loosening which consequently leads to inflammation at the bone–implant interface and, in some cases, leads to periprosthetic fractures. These complications may lead to revision surgeries where the success rate is substantially lower, reported to be 35% at 10 years follow-up [8]. Compared to primary hip replacement surgery, the cost of revision surgery is significantly higher due to the complexity associated with revision technique [9]. Therefore, osteoporosis treatment imposes a significant financial burden on healthcare systems and poses a significant threat to the quality of life and survival of elderly patients.

The complications associated with osteoporosis and total hip arthroplasty have been highlighted in this review in three sections related to biomechanical properties, implant optimization and drug-laden hydrogels, see Figure 1. This review aims to identify and discuss articles introducing and investigating treatment solutions for osteoporotic patients submitted for total hip replacement. Studies obtained from the literature have assessed biomechanical properties and structure of osteoporotic bones, hip implant optimisations, and drug, cells, and bio-activators delivery through hydrogels.

## 2. Materials and Methods

The search was conducted using three search engines: Scopus, PubMed, and Web of Science in April 2022. The method implemented in the search was (Title–Abstract–Keywords) as follows: TITLE-ABS-KEY (“Osteoporosis” AND (“hip implant” OR “hip replacement” OR “joint arthroplasty” OR “joint replacement”) AND (“biomechanics” OR “tissue engineering” OR “regenerative medicine” OR “bone implant” OR “tissue scaffold*” OR “hydrogel” OR “modelling” OR “modeling”)).

All potential articles generated by the search engines were screened by the title and abstract and were subjected to pre-set inclusion and exclusion criteria. The inclusion criteria for this review were: (a) articles on hip replacements and total hip arthroplasty as a consequence or with the presence of osteoporosis, (b) studies conducted for the purpose of evaluation of existing diagnostic techniques or the establishment if new ones. The exclusion criteria were: (a) studies focused on drug treatments rather than engineering or surgical interventions, (b) studies focused on the immune system, (c) studies on any skeletal parts other than hip, (d) studies in any language other than English, and (e) any document type other than original articles and reviews.

## 3. Results

The search engines used in this review produced 183 potential sources: 104 by Scopus, 18 by PubMed, and 61 by Web of Science. There were 45 duplicates, 94 excluded by abstract screening, and 15 excluded by reading the full text. Therefore, 29 papers were included in this review. Of those 29 papers there were 25 representing current diagnostic and treatments approaches and 4 articles introduced novel treatment approaches.

### 3.1. Biomechanical Properties and Microstructure

The asymptomatic nature of osteoporosis encouraged researchers to utilise and assess tools that help characterising this disease to guide treatment approaches. In this review, the results include studies investigating accuracy of diagnostic devices, and the role of mechanical tests and finite element analysis (FEA) in the assessment of osteoporotic bones. These three methods of evaluation are complementary to one another. It is possible to develop FE models with the assistance of diagnostic tools, and FEA can be validated with the assistance of mechanical testing.

#### 3.1.1. Diagnostic Tools

The results obtained from these diagnostic devices were interpreted to understand the characteristics of osteoporosis in general and its behaviour with different variables such as age and gender. The tools varied regarding data acquisition, and accuracy as illustrated in Table 1. Osteoporotic bones can be assessed by analysing the femoral neck on the unaffected side of a simple anterior-posterior X-ray, the severity of osteoporosis can be classified into one of six grades, referred to as Singh Index as illustrated in Figure 2. This index attributable to the rarefaction of trabecular structures [10]. Although this tool was reported to be an inexpensive approach for bone architecture assessment, the assessment acquired by Singh Index would be an estimation rather than accurate [11], whereas Singh Index value was combined with bone mineral density (BMD) evaluation and reported as an acceptable approach to investigate the mechanical competence of bone [12]. In fact, BMD has also been used in combination with another assessment techniques such as velocity ultrasound, and that combination was reported to improve the fracture risk assessment for osteoporotic patients [13]. It was also suggested by Endo et al. [14], that the assessment of osteoporotic bones using magnetic resonance imaging (MRI) could enhance the accuracy of the assessment conducted using BMD only. The value MRI can add to BMD is that it can predict the strength of cancellous bone in addition to the bone quality change [14]. It is worth mentioning that BMD can be assessed using dual energy X-ray absorptiometry (DEXA) which was reported to be the most accurate and reliable technique for the assessment of BMD [15].

**Table 1 bioengineering-10-00161-t001:** Diagnostic tools used for bone assessment and their efficiency as reported.

Tool	Use	Results	Reference
Singh Index (SI)	Bone architecture assessment	Inexpensive tool, but not accurate results	[11]
Singh Index (SI) + Bone Mineral Density (BMD)	Mechanical competence and architecture of the bone	Acceptable estimation compared to Singh Index alone	[12]
Velocity Ultrasound + Bone Mineral Density (BMD)	Fracture risk assessment	Improved in comparison with Singh Index alone	[13]
Dual-Energy X-ray Absorptiometry (DEXA)	Evaluate (BMD)	Excellent for the assessment of (BMD)	[15]
Magnetic Resonance Imaging (MRI)	Evaluate (BMD)	Enhance accuracy of (DEXA) results	[14]
Low Field Nuclear Magnetic Resonance (LF-NMR), High Resolution Computed Tomography (HR-CT), and micro-CT (μCT)	Evaluate bone porosity and structure	Qualitative and quantitative information that can be used for Finite Elements Analysis	[16]

**Figure 2 bioengineering-10-00161-f002:**
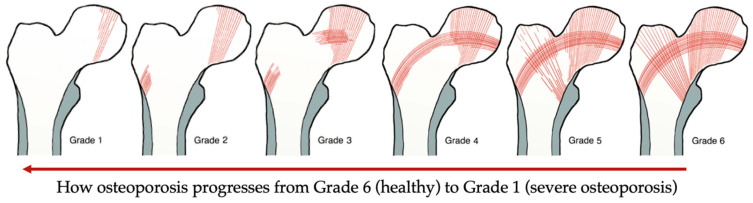
Singh Index grades: Grade 6 The radiograph clearly shows each of the trabecular subgroups. Cancellous bone appears to fill the whole top of the femur. Grade 5: the primary tensile trabecula has been highlighted, and the Ward triangle is clearly visible. Grade 4: the primary tensile trabeculae are significantly diminished, but can still be traced from the lateral cortex to the upper femoral neck. Grade 3: the continuity of the major tensile trabeculae is broken. Grade 2: only the major compressive trabeculae are visible, but other groups have been assimilated. Grade 1: the number and size of the main compressive trabeculae are diminished and no longer prominent. Adapted from reference [17].

Another view to consider is the suggestion, made by Porrelli et al. [16], that morpho-logical information cannot be extracted from DEXA and ultrasonography alone in a qualitative and quantitative manner. It was reported that using a combination of MRI, low field nuclear magnetic resonance (LF-NMR), high resolution computed tomography (HR-CT), and micro-computed tomography (µCT) have enhanced the study of bone porosity and structure. The reason for classifying these techniques as accurate and more informative is due to the ability to build models based on the obtained data for FEA [16].

#### 3.1.2. Mechanical Testing

Mechanical testing in the field of tissue engineering can be conducted for various reasons such as bone stress, strain, stiffness, failure load, and fracture risk assessment as shown in Table 2.

In a biomechanical study investigating the bone fragility and mechanical behaviour, compression testing has revealed that men have lower fracture risk compared to women in the presence of osteoporosis in both populations [18]. The findings of this study indicate that gender is one of the variables which must be taken into account while considering a treatment plan for an osteoporotic patient.

Mechanical testing can also be used to assess load bearing with the presence of fractures in addition to the mechanical evaluation of different fixation approaches [19,20]. On a total of six osteoporotic female cadaveric pelvises, Marmor et al. [19] produced posterior wall fractures. After the fracture was created, cyclic loading equal to 1.8 times the body weight was applied. Every specimen was able to withstand the loading with a cup motion of less than 150 µm, which is within the permissible limit. Similarly, Jenkins et al. [20] tested the fracture toughness for three groups: osteoporosis, osteoarthritis, and control group. It appeared that neither osteoporosis nor osteoarthritis have any additional influence on the fracture toughness of the inferomedial femoral neck beyond that which is caused by natural ageing.

In research that aims to create therapeutic options for osteoporosis, it is crucial to possess bone samples that represent this skeletal condition in order to examine and evaluate the approach. Gluek et al. [21] have introduced and evaluated a novel synthetic bone with a mechanical reaction equivalent to that of osteoporotic bone from cadavers.

#### 3.1.3. Finite Element Analysis

Medical engineering has implemented FEA in studies of bone structure, mechanical properties, and assessments of treatment approaches as shown in Table 3. The accuracy of the data obtained by FEA primarily depends on the CT scans from which the models are constructed [22]. Rieger et al. [23] stated that their approach to study and assess bone macrostructure and microstructure has also been used by a number of scholars. They used high-resolution µCT images of fractured femoral heads to produce µFE mesh in order to obtain bone stress and strain. They stated that the mechanical properties of the bone on the macroscopic level can be obtained by the analysis of the microstructure only. Their findings showed that using FEA in addition to numerical calculations based on that FEA as an inversed approach can reveal macroscopic and microscopic mechanical properties of the bone as they reported their results indicating osteoporotic bones have comparable elasticity to healthy ones. However, the only difference identified was the yield stress with a mean of 85.6 ± 16.7 MPa which is lower than yield stress of healthy bones. It is worth mentioning that this approach was suggested to add supportive data to the histomorphometric analysis in the orthopaedic studies.

Similarly, He et al. [24] implemented FEA to compare and understand osteoporosis and osteoarthritis by analysing bone structure and mechanical behaviour. In their study, the bone microstructure was generated through virtual biopsies obtained from µCT scans of the subchondral trabecular bone. The FEA results showed that the plate and rod structures are significantly higher in the osteoarthritis group compared to the osteoporosis group, which consequently the failure load, stiffness, young’s modulus, compressive strength, yield strength, and maximum compressive force are reported to be higher in the osteoarthritis group.

### 3.2. Implant Optimisation

The use of hip prothesis has been a huge leap in the treatment of skeletal diseases, as it was stated that primary total hip replacement (THR) conveys more desirable outcomes as a treatment intervention in comparison to other approaches such as open reduction internal fixation regarding the stability of the acetabular component especially for osteoporotic patients [25]. However, the complications associated with this procedure opened an area of research for the purposes of ensuring success in the long-term. Since the major downside of using an artificial hip joint, has been stem instability which consequently leads to further complications. There have been different approaches reported in the literature to enhance implant fixation and long-term stability such as design, and surface finish optimisation. FEA has also been used to investigate factors that lead to complications, the results obtained from these investigations has provided some insights that inspired implant optimisation.

#### 3.2.1. Design Optimisation

Altered implant designs compared to conventional ones were implemented in clinical trials to eliminate aseptic loosening and periprosthetic fractures (Table 4). The implant stem was shortened to be used in THA for osteoporotic patients [26,27]. It was reported that a short, tapered stem can show desirable stability compared to conventional; refer to Figure 3, which shows both short and conventional implants in a scan.

Although the success rate in Santori et al.’s [26] clinical trial with regards to aseptic loosening was reported to be 100%, there are some cases were periprosthetic fractures occurred; whereas, Zhen et al. [27] reported that the utilisation of short-stem hip joint has eliminated both aseptic loosing and, periprosthetic fractures, and thigh pain, yet their study had some limitations which may influenced their conclusions. The mean duration of the follow-up after operation was (5.5 ± 1.1 year) which was deemed short and insufficient, in addition to the low number of patients.

Implant design alteration was also used to tackle femoral neck fractures (FNFs) phenomena. An approach of using dual-mobility cup in total hip arthroplasty (DM-THA) procedure was conducted and evaluated on osteoporotic Chinese population [28]. The use of DM-THA has shown desirable outcomes regarding dislocation of FNFs, yet in their clinical study there was a need for revision due to loosening. These design manipulations have shown solutions for some of the complications associated with osteoporosis such as acetabular component fixation; however, periprosthetic fractures, and aseptic loosening still existed with those designs. Therefore, scholars have been investigating the effectiveness of implant surface treatment to enhance fixation and stability.

#### 3.2.2. Surface Finish Optimisation

Large area electron beam melting (LAEB) was used to adjust the nanotopography of the titanium alloy surface used for joint implants. The resultant surface roughness with topography Ra of ~40 nm was reported to enhance the osteogenic differentiation in vitro on human skeletal stem cells (SSCs) [29]. However, the mechanical properties, mineralisation, and the bone matrix organisation of an implant treated with LAEB have not been investigated in vivo.

#### 3.2.3. Finite Element Analysis

Conducting FEA for the purpose of anticipating the success rate of hip joint implants was conducted by Rafiq et al. [30] to assess the feasibility of using a cementless implant for osteoporotic patients. The FE algorithm used was simulating stairs-climbing to investigate micromotion at the bone–implant interface. An osteoporotic model was compared to healthy and osteoarthritic models. Poor bone density, stiffness and thin cortical bone in the osteoporotic model allowed an increase in the surface area which compromised bone growth and implant stability observed by micromotion. The analysis findings suggested that cementless implants are predicted to experience loosening on the long-term with osteoporotic host bone.

### 3.3. Drugs, Cells, and Bioactivators

Some studies focused on the effect that osteoporosis has on hip prothesis after the implantation and how that can be reversed by using anti-osteoporosis drugs, stem cells and bio-activators for the purposes of restoring the natural bone remodelling process which is compromised by osteoporosis. These substances have been investigated when delivered orally or as an implant coating. Bone grafts were also investigated for their desirable bioactivity. Yet, the most recent approach found in the literature for drug delivery into bones is hydrogels.

#### 3.3.1. Implant Coating

Several studies have investigated the effect of implant surface treatment on the stability of the bone–implant interface as illustrated in Table 5.

It is worth noting that bisphosphonates were reported to increase the fatigue life on bone cement when they are mixed in a powder form [35]. However, different forms of bisphosphonates were compared to Zoledronate, and the results showed that Zoledronate demonstrated desirable enhancement of early bone formation and bone–implant integration, as shown in Figure 4. Gao et al. [32] stated that the benefit of bisphosphonate immersion on the implant surface is that they impact osteoclasts by eradicating their proliferation activity, which is desirable for patients with osteoporosis. Another study suggested that the use of hydroxyapatite-coated implants enhances bone mineralisation, formation, and mechanical stability; however, using such implants may lead to loosening and inflammation in the implant site due to the damage it causes to the bone matrix [33].

Another approach reported to have a potential to eliminate osteoporotic effects and influence THA success which is using calcium phosphates (CaP) to coat the implants [34]. It is worth noting that the CaP-coated implant was investigated along with platelet-rich plasma (PRP) treatment, which may compromise the accuracy of the conclusions made about the effects of CaP coating on its own.

#### 3.3.2. Bone Grafts

The use of bone autografts and allografts in primary and revision arthroplasty have been well established as a surgical solution [36]. It was also stated by Brewster et al. [36] that the purpose of implementing a bone graft is to enhance the cup stability and fixation in the hip, due to their superior ability to withstand complex forces (i.e., normal loads, and shear strain). They noted that osteoporotic specimens such as femoral heads can provide similar properties to healthy ones regarding bone grafts, though with much fewer particles.

#### 3.3.3. Hydrogels within Metallic Scaffolds

Several scholars have studied the feasibility of using hydrogels as drug delivery vehicles for osteoporotic bones owing to their desirable biocompatibility in addition to their ability to provide a medium where impregnated cells can survive during the desired course of release Table 6 [37,38,39,40]. Hydrogels have been studied within metallic scaffolds as a treatment approach in fracture site for enhancement of bone remodelling and osteointegration processes.

Hydrogels were preliminarily investigated in vitro and in vivo to evaluate their potential contribution in the development of orthopaedic complications solutions.

The four studies identified in the literature that implemented impregnated hydrogels all used hydrogels in combination with porous 3D printed titanium scaffolds to investigate their effect on osteoporotic bones. They assessed the biocompatibility, cell proliferation, cell differentiation, and mechanical stability of the composite implants.

The hydrogels in those studies were used as drug and cells delivery vehicles, they were impregnated autophagy-regulated rapamycin [37], bone morphogenetic protein-2 (BMP-2) [38,40], and technetium methylenediphosphonate (T99c-MDP) [39]. In fact, Bai et al. [40] have also impregnated bone marrow stem cells (BMSCs) in the hydrogels with the BMP-2. They were investigated in three groups to distinguish the effect of each compared to the effect of them acting together.

It is crucial to evaluate the biocompatibility of the hydrogels when they are being investigated for biological applications, to understand their effect in promoting cell adhesion and proliferation. In all four studies, it was reported that the hydrogels show good biocompatibility, cell proliferation, and cell differentiation using biological indicators (i.e., Calcein acetoxymethyl ester (AM)/propidium iodide (PI) staining, and Alizarin red staining). Although, Bai et al. [40] reported that there were negligible inflammations after implantation, but the bone–implant site gradually became normal along the course of the study. This inflammatory reaction by the host bone towards the implanted composite scaffolds were not observed in the other studies.

Wang et al. [38] and Cui et al. [39] reported that the gel was formed using a thermosensitive approach. Poloxamer 407 was used as a powder to be added into a solution of sterilised phosphate-buffered saline (PBS)and kept at 4 °C until the solution was transparent. The drug investigated in the studies was added to the solution at the same temperature of 4 °C, the mixtures were subjected to a temperature of 37 °C until gelation was achieved.

However, in Li et al. [37] and Bai et al. [40] chemical and crosslinking approaches were used for the gel formation process. Strong hydrogen bonds between polyvinyl alcohol (PVA), N-carboxyethyl chitosan (NCECS), and agarose (AG) in the form of NCECS-AG, NCECSPVA and AG-PVA solutions, were the main factor of fabricating the hydrogels in Li et al. [37], where gelation was instantly achieved by the intended chemical reaction; whereas, an in-situ crosslinking approach was performed to prepare the solution of N-carboxyethyl chitosan (N-chitosan) and adipic acid dihydraside (ADH) with hyaluronic acid-aldehyde (HA-ALD) in Bai et al. [40]. The hydrogel was formed using a Lab Dancer to achieve homogeneity.

The degradation rate varied in each study which indicated that different materials and preparation methods could alter the degradation process. Li et al. [37] reported that the hydrogels have a slow degradation rate where the process took 36 days to degrade in vitro almost completely. Whereas, in Wang et al. [38] the drug release profiles were observed over the course of 20 days by which 70% of the of the drug was released, and that was a result of both drug diffusion and hydrogel degradation. It was also reported in Wang et al. [38] that due to protein concentrations, the detection of the drug was difficult, therefore, only 70% of the degradation was detected in that study. In Cui et al. [39] and Bai et al. [40], the hydrogels were reported to completely degrade in 12 and 28 days, respectively. It is noted that thermosensitive hydrogels degrade at a faster rate compared to the ones fabricated via crosslinking.

The four studies have observed the microstructure of the hydrogels after the gelation processes, and that was conducted using scanning electron microscope (SEM). The pore size of the hydrogel was reported to be approximately between 100–200 µm which is favourable to provide space for osseointegration where desired, on the scaffold interface, and inside the pores of metallic scaffold. The advantage of allowing bone formation inside the pores of the scaffold is increasing its stability and attachment with the host bone.

One of the outcomes of the in vivo study conducted in Bai et al. [40] is that impregnating the hydrogels with a combination of BMSCs, and BMP-2 have improved the bone regeneration process compared to the case of implementing the porous scaffold alone as shown in Figure 5. The results obtained by Bai et al. [40] have also revealed that the mechanical properties were significantly increased in the implant impregnated both BMSCs and BMP-2 compared to unfilled porous titanium scaffold (*p* < 0.01), evident by the peak values of the push-out tests. The desirable enhancement in the mechanical properties was achieved by the high level of osteointegration formed between the implanted scaffold and the host bone.

Similarly, Li et al. [37], Wang et al. [38], and Cui et al. [39] performed push-out tests and their results indicated higher peak values when porous titanium scaffolds were filled with impregnated hydrogels (*p* < 0.01 or *p* < 0.001). Although each study investigated different substance impregnated in hydrogels, they have comparable osteointegration against titanium scaffolds on their own.

Remarkably, Cui et al. [39] and Wang et al. [38] investigated the osteoclastic activity and found that both osteoprotegerin (OPG) and technetium methylenediphosphonate (T99c-MDP) inhibit osteoclasts proliferation evident by RANKL expression (*p* < 0.001 or *p* < 0.0001), respectively, compared to the pure titanium scaffolds.

On the other hand, Li et al. [37] investigated the bacterial proliferation by examining the absorbance of *S. aureus* and MRSA. It was reported that the silver nanowires (Ag-NWs) have significant effect that inhibited bacterial proliferation, which means that Ag-NWs can eliminate postoperative inflammations. It was also observed that when porous scaffolds were not filled with hydrogels, bacterial proliferation was greater than in the control group, which indicate grater bacterial reproduction allowed in the voids of the porous structure.

## 4. Discussion

The asymptomatic nature of osteoporosis increases the chance of bone fracture occurrence due to fragility. Therefore, diagnostic tools are vital for preventive and analytical purposes. BMD has been classified as one of the most important markers of bone quality, and the most effective way to assess BMD is using DEXA scans [16]. It was also found that there have been studies implemented and validated using FEA as an assessment and predictive tool of bone quality and behaviour [23,24]. The understanding of osteoporosis led to attempts to overcome complications regarding fractures, in fact, these attempts were even extended towards the improvement of the device that had been used for osteoporotic patients (i.e., total hip replacement). Although implant design optimisations were reported to improve the postoperative loosening, periprosthetic fractures still existed [26]. These unfavourable results may be attributable to the fact that osteoporosis is a degenerative illness, indicating that optimisation of the implant design may achieve short-term success but will not provide long-term stability.

On the other hand, bisphosphonates, alendronates in particular, have been widely used in daily oral doses as a treatment for osteoporotic patients and shown good results regarding the enhancement of bone remodelling [41,42,43]. Therefore, the desirable outcomes of oral bisphosphonates treatments encouraged scholars to immerse implants with bisphosphonates as a targeted treatment to increase the stability and fixation of the implant, and that was done using Zoledronate [31,32]. Immersing Zoledronate on the surface of the implant showed higher osteointegration on the bone–implant interface compared to oral dosing of alendronate. It was noted that implants immersed with bisphosphonates were synthesised using titanium alloys and coated with hydroxyapatite before immersion. However, the effect of hydroxyapatite coating was reported to deteriorate bone–implant ingrowth in osteoporotic patients [32,33]. Although they reported that there are undesired effects of hydroxyapatite, their study groups were coated before immersion. Their practice indicates that one cannot be used without the other.

Despite the desirable properties of bone grafts in promoting cup stability and fixation and their biocompatibility, it was reported that the use of a bone grafts has a high infection rate in addition to its effect of blocking revascularisation with the close packing, and rare restoration of muscle attachment [36,44]. Further, in bone remodelling, grafts may degrade within this process in which stability would be compromised [36]. In addition to their limited supply, the use of autografts is not practical for osteoporotic patients due to biomechanical complications associated with osteoporosis.

In studies that investigated hydrogels as treatment approaches with orthopaedic devices, the results are promising and they address the current complications associated with THR [37,38,39,40]. As stated earlier, osteoporosis compromises the success rate of THR by the low osseointegration of the bone–implant interface which may lead to aseptic loosening, severe inflammation, secondary fractures, and consequently revision surgeries. Those complications were eliminated by implanting porous metallic scaffolds loaded with hydrogels impregnated with cells, drugs, and growth factors, into osteoporotic bones in vivo. The composite scaffolds with impregnated hydrogels have conveyed significant increases in osteointegration and blocked the undesired osteoclastic activity which causes the excessive bone resorption. The composites have also been impregnated with sliver nanowires which promoted antibacterial activity which inhibited bacterial proliferation and inflammation. Li et al. [37], Wang et al. [38], Cui et al. [39], and Bai et al. [40] investigated hydrogels within the porous metallic scaffolds therefore, mechanical tests were performed to assess osteo-integration on the bone–implant interface.

The promising results reported by Li et al. [37], Wang et al. [38], Cui et al. [39], and Bai et al. [40] of using impregnated hydrogels are a huge leap in the treatment of osteoporotic bones. The significance of this treatment approach is that the impregnated substances work on restoring the biological activities affected by osteoporosis such as the lack of bone ingrowth and osteointegration, and the excessive bone resorption. Yet, these studies neither assessed the mechanical behaviour of the hydrogels nor did they investigate the effect of impregnated hydrogels beyond metallic scaffolds. The influence of those impregnated hydrogels could have a positive effect on osteoporotic bone and form a composite with the native bone similar to the presented composite with metallic scaffolds. The results of such investigations may lead to a preventive treatment approach for osteoporotic patients before a fracture occurs.

## 5. Conclusions

In this review, it was found that there have been attempts to overcome complications associated with osteoporosis when patients are submitted for total hip replacement due to fragility fractures. The optimisation of the implant design was reported to show short-term stability. However, the degenerative nature of osteoporosis has led to loosening and periprosthetic fractures in the long term. However, stability was improved when bisphosphonates were used on the implant surface to target the implantation site and reverse the osteoporotic effect. Their effect was compromised by the presence of hydroxyapatite which deteriorate bone ingrowth in osteoporotic patients.

Promising results were found in studies that used hydrogels impregnated with cells, drugs, and growth factors within metallic scaffolds. Significant increase in osteointegration was observed in addition to the inhabitation of osteoclastic and bacterial activities. This is an indication that this approach restores biological activities compromised by osteoporosis.

The results from the use of impregnated hydrogels for osteoporosis are promising to improve osteointegration and block excessive osteoclastic activity. This suggests that hydrogels merit further investigation, which could include their use for preventative strategies as well as investigating further novel approaches to improve the outcomes for total hip replacements.

## Figures and Tables

**Figure 1 bioengineering-10-00161-f001:**
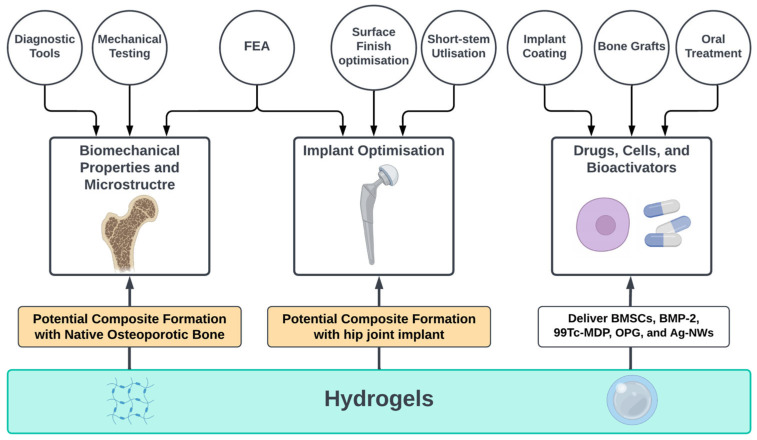
Overview of the review highlighting the importance of osteoporotic bone biomechanics, implant optimisation and drug delivery systems, created with BioRender.com (accessed on 9 January 2023).

**Figure 3 bioengineering-10-00161-f003:**
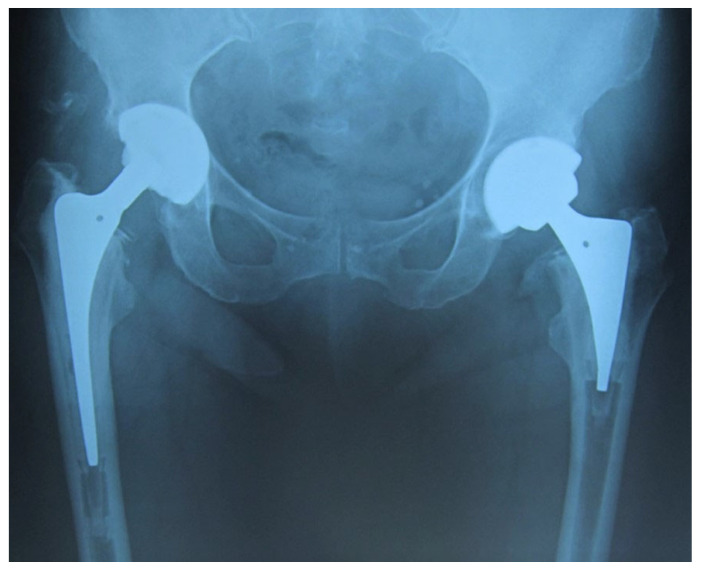
A patient with both conventional implant (Left), and short stem implant (Right) [26].

**Figure 4 bioengineering-10-00161-f004:**
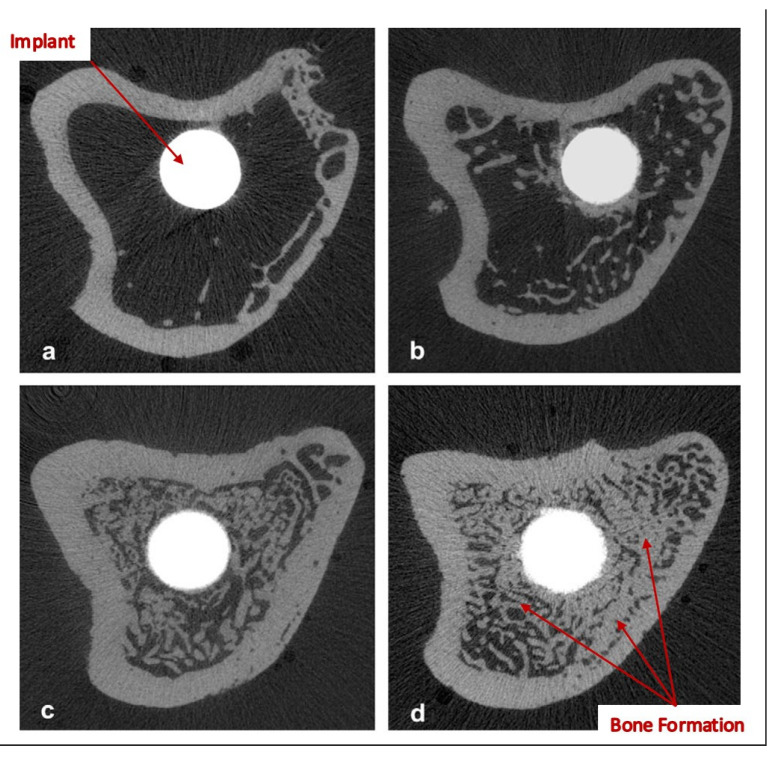
Micro-CT binary images of tibiae with implants 3 months after implantation: (**a**) uncoated implant; (**b**) Pamidronate; (**c**) Ibandronate; (**d**) Zoledronate [32].

**Figure 5 bioengineering-10-00161-f005:**
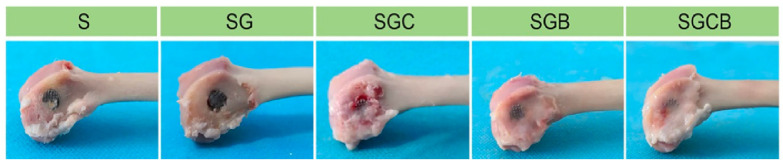
Bone regeneration at the implant interface. (S) scaffold, (SG) scaffold with hydrogels, (SGC) scaffold with bone marrow stem cells (BMSCs) impregnated into hydrogels, (SGB) scaffolds with bone morphogenetic protein-2 (BMP-2) impregnated hydrogels, and (SGCB) scaffolds with BMSCs and BMP-2 impregnated into hydrogels. Figure adapted from reference [40].

**Table 2 bioengineering-10-00161-t002:** Mechanical testing approaches.

Aim	Type of Test	Results	Reference
Determine gender effect on fracture risk	Compression	Males have a bone Young’s modulus of 293.68 MPa and an ultimate stress of 8.04 MPa, whereas females have 174.26 MPa and 4.46 MPa for young’s modulus and ultimate stress, respectively. Therefore, men have lower fracture risk compared to women.	[18]
Evaluate the weightbearing immediately after fixation of posterior wall (PW) fractures	Cyclic loading	With assistance, immediate load bearing is allowable with 50% of PW and 25% of acetabular rim, regardless of PW fixation.	[19]
Investigate effect osteoporosis on bone fracture toughness	Fracture toughness	Fracture toughness decreased with ageing (7.0% each decade, r = −0.36, *p* = 0.029), while comparable fracture resistance properties were found in osteoporotic, osteoarthritic and control groups (10% difference for indentation and *p* > 0.05 for fracture properties).	[20]
Introduce synthetic bone that represent osteoporotic cadaveric bones.	Four-point bending, axial compression, and pullout	There was good correlation found between the cadaveric and synthetic bone samples. The *p*-values in all mechanical tests were acceptable, ranging between 0.1–0.9 except in pullout tests (*p* = 0.005).	[21]

**Table 3 bioengineering-10-00161-t003:** FEA for bone microstructure assessment.

Aim	Bone Model	Software	Results	Reference
Evaluate macroscopic mechanical properties the bone	Virtual trabecular bone biopsy from CT scan	Abaqus 6.9-2	Osteoporotic bones have comparable elasticity to healthy ones, with young’s modulus mean (±SD) of 18.92 ± 5.43 GPa. However, the yield stress was found to be lower in osteoporotic bones with a mean (±SD) of (85.6 ± 16.7 MPa).	[23]
Evaluate the influence of plate and rod in osteoporotic and osteoarthritic patients	Virtual subchondral trabecular bone biopsy from CT scan	Scanco Medical Finite Element Software 1.06	Osteoarthritic subchondral bones had higher stiffness with a mean (±SD) of 12,003.56 (±7590.42) kN/mm, while the mean stiffness of osteoporotic bones was 4964.01 (±3778.37) kN/mm. Similarly, the failure load was reported to be higher in osteoarthritic bones compared to osteoporotic ones with 477.7 (±279.56) MPa and 215.89 (±143.73), respectively.	[24]

**Table 4 bioengineering-10-00161-t004:** Implant design optimisation.

Implant Design	Targeted Complication	Results	Limitations	Reference
Short stem implant	Aseptic loosening	The mean of the Harris Hip Score (HHS) in the two groups increased from 45.0 ± 16 (29–61) and 40.0± 11 (29–51) prior to surgery, to 93 ± 9 (84–100) and 96 ± 7 (89–100), respectively. The survival rate with stem revision for aseptic loosening was 100%.	Some cases with Vancouver B1 and Vancouver B2 fractures were reported in both groups.	[26]
Cementless short metaphyseal fitting stem	implant instability	The mean HHS improved from 48.0 ± 8.0(38.0–61.0) prior to surgery to 91.0 ± 8.0 (85.0–98.0). In addition, there were no postoperative complications such as infection, deep vein thrombosis, loosening, or peri-prosthetic fracture.	Low number of patients, and short follow-up duration.	[27]
Dual-mobility cups in total hip arthroplasty (DM-THA)	Femoral Neck Fractures (FNFs)	The mean HHS increased from 58.62 (+15.79) preopratively to 86.13 (+9.92).	Cases of loosening, revision DM-THA, intra-prosthetic dislocation, migration, tilting, and severe wear were reported in the study.	[28]

**Table 5 bioengineering-10-00161-t005:** Implant coating.

Treatment	Targeted Complication	Results	Limitation	Reference
Zoledronate	Instability and poor bone formation in the bone–implant interface	Bone formation was enhanced by the elimination of osteoclastic activity by Zolendronate. Thus, in comparison to the implant not coated with Zoledronate, coated implants showed significantly higher maximal pullout force (*p* < 0.05) and (*p* < 0.01).	Used with hydroxyapatite coating, which is reported to impair osteoporotic bone ingrowth, consequently long-term survival.	[31,32]
Hydroxyapatite (HA) coated implants	Poor bone–implant ingrowth	The mean osseointegrated implant surface (OIS) in implants coated with HA and uncoated ones were 23.7 and 23.5 in ovariectomised rats, respectively. HA have no effect on osteoporotic bones while it enhances the OIS in healthy bones.	The results of the study indicate that HA-coated implants deteriorate bone ingrowth in the long term.	[33]
Calcium Phosphates coating (CaP) with platelet-rich plasma (PRP)	Implant instability	Enhanced stability, evident by the increase in the maximal push-out force in the group treated with CaP and PRP compared to the control group (*p* < 0.05).	No limitations mentioned in the study	[34]
Surface Large Area Electron Beam melting (LAEB)	Implant surface nanotopography	When compared to the untreated group, the group treated with a cathode voltage of 35 kV and 25 shots showed a significant increase in osteogenic activity (two- to three-fold). This peak was observed to correlate with a surface roughness ($R_{a}$) of 44 nm.	The technique has not been investigated in vivo for mechanical interface strength	[29]

**Table 6 bioengineering-10-00161-t006:** Hydrogels within metallic scaffolds.

Material	Fabrication Process	Impregnated Drugs	Reference
NCECS-PVA and AGPVA	Chemical crosslinking	Autophagy-regulated rapamycin	[37]
Poloxamer 407	Thermosensitive mixture	Bone morphogenetic protein-2 (BMP-2)	[38]
Poloxamer 407	Thermosensitive mixture	Technetium methylenediphosphonate (T99c-MDP)	[39]
N-carboxyethyl chitosan (N-chitosan)	In situ crosslinking	Bone marrow stem cells (BMSCs) + (BMP-2)	[40]

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
