# Peer review of "A Review of Conventional and Novel Treatments for Osteoporotic Hip Replacements"

_bioengineering, 2023, doi:10.3390/bioengineering10020161_

Round 1
Reviewer 1 Report
The article is well written and covered the topics.
In the table 2,3,4 and 5, the results of the articles are summarized in short sentences. I recommend to add representable data such as odds ratio or how much reduced the risk of fracture to the table. That will make the readers to compare each treatments easily.
Reviewer 2 Report
It was a review study about the application of different methods for the treatment methods used for the osteoporotic Hip replacements. Here are some comments related to this study that should be considered before publication:
1- Please mention the references of tables in their last column.
2- “the mechanical evaluation of different fixation approaches [19], [20].” Please write references like this [19, 20]. The same for “bone morphogenetic protein-2 (BMP-2) [38], [40], and”. “dictive tool of bone quality and behaviour [23], [24].” And others in discussion.
3- “Gao et. al. [32] stated that the advantage of immersing bisphosphonates on the implant surface as is their effect of blocking osteoclasts by eliminating their proliferation and activity which is highly desirable for osteoporotic patients.” please rewrite.
4- “The four studies identified in the literature used porous 3D printed titanium scaffolds soaked into hydrogels, then tested their biocompatibility, cell proliferation, cell differentiation, and mechanical stability of the scaffold within the bone.” please rewrite.
